# Unravelling the French National Fungal Database: Geography, Temporality, Taxonomy and Ecology of the Recorded Diversity

**DOI:** 10.3390/jof8090926

**Published:** 2022-08-31

**Authors:** Montan Gautier, Pierre-Arthur Moreau, Béatrice Boury, Franck Richard

**Affiliations:** 1Centre d’Ecologie Fonctionelle et Evolutive (UMR CEFE), University Montpellier-CNRS-EPHE-IRD, 1919 route de Mende, CEDEX 5, F-34293 Montpellier, France; 2Laboratoire de Génie Civil et géo-Environnement (ULR 4515-LGCgE), University Lille, F-59000 Lille, France; 3Association pour le développement d’outils naturalistes et informatiques pour la Fonge (AdoniF), 3 rue du Pr Laguesse, F-59000 Lille, France

**Keywords:** fungal diversity, large dataset, citizen science, geography of fungal knowledge, mycological societies, fungal ecology, rarity, fruitbodies, species richness

## Abstract

Large datasets are highly valuable resources to investigate multi-scale patterns of organisms, and lay foundations for citizen science-based conservation strategies. Here, we used 1,043,262 records from 1708 to 2021 to explore the geography, taxonomy, ecology and distribution patterns of 11,556 fungal taxa in metropolitan France. Our analysis reveals a four-phase pattern of temporal recording, with a main contribution of post-1977 observations in relation with the structuration of associative mycology. The dataset shows an uneven geography of fungal recording. Four clusters of high-intensity sampling scattered across France contrast with poorly documented areas, including the Mediterranean. Basidiomycota and Agaricales highly dominate the dataset, accounting for 88.8 and 50.4% of records, respectively. The dataset is composed of many rare taxa, with 61.2% of them showing fewer than 100 records, and 20.5% recorded only once. The analysis of metadata brings to light a preponderance of the mycorrhizal guild (44.6%), followed by litter saprotrophs (31.6%) and wood saprotrophs (18.1%). Highly documented forests (76.3% of records) contrast with poorly investigated artificial (6.43%) and open habitats (10.1%). This work provides the first comprehensive overview of fungal diversity in France and identifies the Mediterranean area and open habitats as priorities to integrate into a global strategy for fungal conservation in France.

## 1. Introduction

Since the early 21st, the emergence of large citizen-science projects and the development of big-data computing and analytical methods generated an unparalleled use of large fungal datasets as powerful tools to explore the spatio-temporal occurrence of fungal species at national [1,2,3,4,5,6] and supra-national scales [7,8] (see also [9,10] for molecular database initiatives). Mainly concentrated in the northern parts of the Palearctic and the Nearctic regions, these initiatives poorly concerned continental areas located in the Mediterranean (see [11,12] for few exceptions in the Iberic peninsula), despite the high interest of this area for the conservation of a unique fungal diversity [13,14,15].

During the last decade, this resurgence of interest for old fungal records was essentially driven by an increasing urgency to understand the response of continental biota, including fungi, to the ongoing climate change [7,16]. Based on the analysis of a large corpus of data from Northern Europe and Northern America, large datasets proved particularly useful to enlighten the response of fungal diversity to global warming, including shifts in the (1) fruiting phenology ([1,17], see [18] for a review), (2) composition of communities [16], and (3) fruit body production of edible species [12]. From an ecological perspective, these large dataset-centered approaches provided new insight into our understanding of the determinants of fungal host shifts ([3]; see also [19]). Last, large datasets were recently combined at the supra national scale to provide an adapted support for the investigation of macro-ecological questions [7].

The astonishing diversity of fungi constantly comes in highly patchy communities of organisms at the local scale, with a large proportion of rarely observed taxa [20,21,22,23]. Long suspected to artefactually result from limited sampling efforts [20] which veils the realized communities [24], the high proportion of rare species in fungal communities was confirmed by analyses of mycelia, using the metabarcoding of either soil fractions [25,26] or plant fragments [27,28]. At large scale, fungal diversity patterns remain poorly investigated (see [29,30] for a molecular-based global assessments of soil diversity), despite the increasing availability of large databases [7]. In a recent attempt to investigate the large-scale patterns of species distribution in England and Switzerland, [31] observed a lognormal distribution of fruiting species at large scale, strikingly contrasting with the Pareto distribution of local records assembled in this database. These authors suggest that the citizen-science nature of data may explain this pattern. Further studies using large datasets from other countries are needed to test this hypothesis.

Untangling the distribution patterns of fungal species from the location of mycologists able to grasp this information and the geography of their recording activity across the landscape is particularly challenging [32,33,34]. At the national scale, it was evidenced that the uneven distribution of the population, but also the heterogeneity of landcover and the topology of human infrastructure combine their effects to induce highly biased spatial patterns of observations [6]. However, the geography of fungal data gathered in large databases remains poorly described, and its relationship with the location and the temporal dynamics of the network of mycological associations and fungi field observers has been poorly addressed so far.

In the Palearctic region, initiatives to assemble and analyze large fungal datasets at the national scale took shape in countries from the north-western (Germany, the U.K., Denmark, Norway, Estonia and the Netherlands) and central parts (Austria, Switzerland and Slovenia) of Europe [7]. Contrastingly, the southern part of the continent still fails to provide equivalent overviews of the accumulated corpus of fungal records at national scales. Located at the interface between Mediterranean frontage and the continental domain on one hand, exposed to the Atlantic influence in the western part of its territory on the other hand, France presents a remarkable diversity of climatic influences, resulting in a highly diverse mosaic of natural habitats [35,36]. In France, a long succession of talented mycologists and a long-term dynamic of local mycological associations contributed to gather a two-century long history of fungal records and collections [16,37]. Despite the long-term vitality of French mycology, the first national mycological database was only initiated in 2008 [38,39]. This collaborative project was developed through the involvement of the Société Mycologique de France (SMF), the Société Mycologique du Nord de la France (SMNF), the Muséum National d’Histoire Naturelle (MNHN) and the University of Lille2, with the support of the ministry of the ecological transition and the Nord-Pas-de-Calais region [39]. Hosted by the association ADONIF since 2015, the French national mycological database encompasses three complementary resources, i.e., a database of located fungal records (FongiBase), an associated taxonomic reference tool (FongiRef) and the corresponding bibliography (FongiDoc; [39]). Online since 2017, this database still waits to be handled in a global analysis of the corresponding patterns of fungal diversity in France.

Here, we analyzed the French national mycological database, which represents the current state of the fungal records accumulated and archived in metropolitan France since 1708 [40]. More specifically, we successively questioned (1) the geographical patterns of 104,3262 located fungal records, (2) the temporal sequence of fungal data acquisition by fungi observers, with particular attention to the contribution of mycological associations, (3) the distribution patterns of fungal alpha diversity across the territory, (4) the taxonomic dimension of observations, (5) the structure of the dataset when considering the abundance patterns of taxa, (6) the contributions of fungal ecological guild to the database, and finally (7) the characteristics of fungal records in relation to the FongiBase contributors. Doing so, we aim at providing a complete analytical framework of the fungal diversity and the network of mycological contributors in France.

## 2. Materials and Methods

### 2.1. National Fungal Database

Fungal records deposited in the French public database are organized into three complementary datasets [40] presented hereafter.

A first dataset is composed of 1,100,420 fungal records (including lichenized fungi) registered in FongiBase on 25 March 2022. FongiBase records are systematically transferred to GBIF through the French SINP database (https://inpn.mnhn.fr/informations/sinp/presentation, accessed on 25 July 2022). Each of those inputs corresponds to an individual observation of one or several fungal fruit bodies. The dataset was compiled with data originated from French mycological societies and gathers contributions by independent taxonomists and academic institutes (see Appendix A for a detailed list of contributors and their respective input). This dataset includes records which were validated by Pierre-Arthur Moreau, and taxonomically assigned using current nomenclature. At the first of April 2022, 1,091,602 records (98.31%) were assigned at the species level, 18,590 records (1.7%) were assigned at the genus level, 225 records (0.02%) were assigned at the family level, 2 records were assigned at the class level, and 1 record was assigned at the order level. All records are associated to an observation date, including 999,760 records (90.8 %) accurate to the day, 1,037,854 records (94.3%) to the month, and 1,099,706 records (99.9%) to the year level. Most observations are spatially located, including 466,678 records (42.7%) with GPS coordinates of the sampling site and 365,421 records (32.9%) with the coordinates of the commune as a proxy of the sampling location. Last, FongiBase associates information concerning the ecology of the collection, using indication of the habitat and/or fruiting support for 110,106 records (10.0%).

The second dataset, FongiRef, provides the current taxonomic framework used for naming records at the validation stage, and the corresponding synonymies used since the observation. As described in Boury et al. (2018), Fongibase is coupled with the nomenclatural database FongiRef (fongiref.fongifrance.fr, accessed on 25 March 2022) updated from TaxRef, the French national taxonomic database managed with the French Museum of National History. All taxa cited and analyzed are “retained names” but data include all taxonomic and nomenclatural synonyms recognized by FongiRef/TaxRef. The last dataset, FongiDoc, is a compilation of publications on fungal taxonomy from which bibliographic data referring to observations with location and date were associated to FongiBase.

### 2.2. Assembling of Analysis Matrices

From FongiBase, we excluded 57,158 records related to lichenized fungi to analyze the distribution patterns, taxonomy and ecology of 1,043,262 fungal observations. We assembled a matrix associating for all records, seven attributes. The matrix contains, for all collections, (1) the validated taxonomic assignment of the records, (2) the observation date, (3) the GPS coordinates of collecting site (or the center of the corresponding commune as a proxy), (4) the corresponding ecoregion, (5) the identity of the observer, (6) when available, the surrounding habitat and (7) for each species, the trophic guild of belonging.

The ecological guild of each species was inferred using the FUNGuild database [41], with the FUNGuildR tool [42]. The species which were absent from FUNGuildR database were assigned to the guild of the corresponding genus when unambiguous or discarded from the analysis when uncertainty remained.

When available, the habitat of records was characterized using information provided by the observers concerning the botanical aspect of the surrounding vegetation. The habitat was assigned to a CORINE biotope code at the class level, and, when possible, at the finer subclass level of the three-level hierarchy ([43]; Appendix A).

For each record, the ecoregion of field observation was assigned using an assignment at the department level. Each department was assigned to one ecoregion (Atlantic vs. Continental vs. Mediterranean vs. Montane [44]). For six departments located at the limit of two ecoregions (e.g., Alpes de Haute-Provence, Hautes-Alpes, Alpes-Maritimes, Ariège, Pyrénées-Atlantiques and Hautes-Pyrénées; see Appendix A and Appendix A), the location of the majority of records was used to assign the department to one ecoregion.

### 2.3. Statistical Analyses

All analyses were performed using R software version 4.1.2 and RStudio software 2022.02.0 + 443 version [45,46]. Maps were obtained using IGN map layers (https://geoservices.ign.fr/adminexpress, accessed on 12 April 2022) and R packages ggplot2, rgdal, and ggspatial [47,48,49].

The density of fungal records was calculated at the department level, as a difference with the national density of records, and expressed in percent. In each department, values were obtained by subtracting the number of observations per square kilometer from the number of observations per square kilometer observed at national scale. The surface of the departments and the number of communes per department were obtained using the French Institute on Economics and Statistics [50]. The specific richness of ecoregions was rarified with the R package vegan by the rarefy function [51].

To determine whether the fungal specific richness varied at the department scale depending on the number and the density of records, the number of mycological societies and the number and the density of inhabitants, we performed a generalized linear model (GLM) with negative binomial distribution to process overdispersion with package MASS [52]. This procedure was selected to take into consideration the non-normal distribution of data. In this analysis, the Paris urban area (i.e., Paris—75, Hauts-de-Seine—92, Seine-Saint-Denis—93 and Val-de-Marne—94) was separately analyzed from the rest of France, because of its specific characteristics (high density of population in a highly restricted geographical area). The number and the density of records, the number of mycological societies and the number and the density of inhabitants were the explanatory variables, and specific richness was the explained variable in the model.

To determine whether the observations and specific richness varied depending on ecoregion, we performed a generalized linear model (GLM) with negative binomial distribution to process overdispersion. Ecoregion was an only categorical fixed factor, and the specific richness and number of records were the explained variables in the model. If the ecoregion effect was significant in the model described above, Tukey multiple comparison tests among the means were performed with package “multcomp” [53]. The species abundance distributions (SADs) were fitted with R package “sads” [54], which uses maximum likelihood methods, and compared with nine different models as in [31].

## 3. Results

### 3.1. The French National Dataset

From a territory of 543,950 km^2^, the analyzed database FongiBase revealed the presence of 11,556 taxa out of 1,043,262 archived fungal records (Figure 1 and Table 1). In Europe, this dataset refers to the largest initiative in terms of the inventoried area, and the fourth database in the number of observations, after the Netherlands (2,094,920 records), Germany (1,530,140 records) and the United Kingdom (1,937,526 records; Figure 1).

### 3.2. Spatial Patterns of Fungal Records

The current geographic distribution of records shows an uneven density of data across administrative units (Figure 2). The highest numbers of records were collected in the departments of Pas-de-Calais (78,199 records; +631.1% as compared to the national density), Doubs (75,227 records; +603.3%) and Jura (70,851 records; +562.4%; Appendix A). When considering the density of records, the highest values are registered in the Territoire de Belfort, Doubs and Jura departments, with 18.8, 14.4 and 14.2 records per square kilometer in average, respectively (Table 1 and Appendix A). At the opposite of the distribution range, the lowest numbers of records have been collected in the departments of Indre (109 records; −99%), Lot-et-Garonne (125 records; −98.3%) and Seine-Saint-de-Denis (221 records; −97.9%; Table 1 and Appendix A). The minimums of record density are registered in Indre, Lot-et-Garonne and Gers departments, with 0.016, 0.023 and 0.057 fungal records per square kilometer, respectively (Appendix A).

The geographical distribution of fungal data recording shows four aggregates of departments with high numbers of observations. Two clusters, located in the Atlantic bioclimatic domain, consist respectively of four and five departments (i.e., Nord, Oise, Nord-Pas-de-Calais and Seine-Maritime on one hand, and Charentes, Loire-Atlantic, Mayenne, Vendée and Vienne on the other hand; Appendix A) which account for 17.4% and 15.6% of the number of located records at the department scale, respectively. A third cluster of high recording intensity, located in the Continental domain, includes Doubs, Jura, Haute-Saône and Territoire de Belfort, and accounts for 18.4% of records (Appendix A). Last, a set of five departments crossing Montane and Continental domains and comprising Isère, Loire, Rhône, Savoie and Haute-Savoie, represents 15.2% of located fungal records (Appendix A). In three of these clusters (i.e., northern, western and eastern), taxa accumulation curves tend to saturate, while the Alpine cluster shows an unsaturated sampling (Appendix A).

In all, the French dataset FongiBase contains 492,014 records from 46 departments located in the Atlantic ecoregion, 342,124 records from 32 departments located in the Continental ecoregion, 35,202 records from 11 departments located in the Mediterranean ecoregion and 157,558 records from 9 departments located in the Montane ecoregion (Table 1 and Appendix A). On average, the number of fungal records per department is significantly lower in the Mediterranean (i.e., 3200 ± 2594), as compared to Montane (17,506 ± 18,861.2; *p* = 0.008), Continental (11,404 ± 18,263; *p* = 0.01) and Atlantic (10,696 ± 16,343.1; *p* = 0.01) ecoregions (Table 1). Similarly, Mediterranean departments harbor lower densities of records (i.e., 0.6 ± 0.4 record.km^−2^) than Montane (3.2 ± 3.28; *p* = 0.008), Continental (2.7 ± 4.7; *p* = 0.001) and Atlantic (2.1 ± 2.7; *p* = 0.008) departments (Table 1).

From an administrative perspective, only 11,536 out of 34,836 French communes, i.e., 32.1%, present at least one fungal record in FongiBase (Table 1). At the ecoregion scale, the proportion of sampled communes ranges from 30.6% (in the Atlantic domain) to 46.7% (in the Montane domain) of the total number of communes (Table 1).

### 3.3. Temporal Patterns of Fungal Data Acquisition

FongiBase contains records from 1708 to 2022, with 9 records (0.0009%) related to observations before 1800, 7226 records (0.7%) related to observations between 1800 and 1900, 28,0438 records (26.9%) related to observations between 1900 and 2000 and 754,879 records (72.4%) related to observations since 2000 (Figure 3A). Similarly, the French database progressively completed its spatial coverage from 7 sampling locations (0.059%) recorded before 1800, to 637 sampling locations (6%) recorded between 1800 and 1900, 5059 sampling locations (43.9%) recorded between 1900 and 2000, and 5829 new sampling locations (50.5%) recorded since 2000 (Figure 3A,B).

The temporal dynamics of accumulation of records and sampling locations show a four-steps pattern. Most ancient data, i.e., recorded from 1800 to 1886, consist of sites with consistently few records (1.8 ± 2 observation per site per year; Figure 3C). A second phase, extending from 1887 to 1977, shows a higher recording activity per site (i.e., 6.9 ± 9.7 records per site per year in average; Figure 3C), accompanied by a highly erratic pattern of data acquisition (from 1.10 records per site in 1903 to 67.38 records per site in 1887; Figure 3C). From 1978 to 2006, the temporal dynamics of record acquisition shows a marked increase, to reach a maximum of 38.8 observation per site in 2016 (19.3 ± 9 records per site per year; Figure 3C). The last phase extends from 2007 to 2022 and shows a high number of records per site (25.8 ± 7.8 records per site per year; Figure 3C), accompanied by a stabilization followed by a progressive decrease of sampling effort per site.

### 3.4. Distribution of Fungal Alpha Diversity

In all, 11,556 fungal taxa are recorded in FongiBase (Table 1). Rarefied species richness slightly varies among the four French ecoregions, from 3632.2 fungal taxa in the Atlantic domain, to 4579 taxa in the Montane domain (Table 1). Continental and Mediterranean ecoregions harbor intermediate richness, with 3686.1 and 4400 taxa, respectively (Table 1). At the department level, there is no difference in species richness among ecoregions, with 1274.2, 1400, 1045.5 and 2099.6 taxa in average in the Atlantic, Continental, Mediterranean and Montane domains, respectively (*p* = 0.09 by ANOVA; Table 1).

At the department level, the number and the density of fungal taxa recorded is significantly and positively correlated with the number (z = 3.39; *p* = 0.0007 by GLM; Table 2) and the density (z = 3.12; *p* = 0.0018 by GLM; Table 2) of fungal records.

At this scale, the number of inhabitants per department is positively correlated to the species richness (*z* = 2.29; *p* = 0.02 by GLM; Table 2). Last, the positive effect of the number of records per department on the number of taxa decreases as the number of inhabitants increases (z = 1.98; *p* = 0.04 by GLM; Table 2).

### 3.5. Taxonomy of Fungal Records

Taxa in the Basidiomycota account for 88.8% (905,400 records) of the recorded observation in the French National fungal database (Figure 4). Ascomycota comes second, with 7.8% (81,622 records) of the total number of observations (Figure 4). Two other fungal lineages, i.e., Zygomycota (0.009%; 98 records) and Chytridiomycota (0.001%; 12 records), account for very few in the database (Figure 4). Last, Fongibase includes few data of groups traditionally observed by field mycologists, including Myxomycota (0.3%; 2955 records), Oomycota (0.004%; 40 records) and Microsporidia (0.00001%; 1 record). Last, 53,109 data (5.1%) cannot be satisfactorily assigned to a correct taxonomic position for current limits in the taxonomic base FongiRef.

At the lower taxonomic level, Agaricales is the most represented order (50.4%), followed by Russulales (12.1%), Polyporales (8.6%) and Boletales (6%; Figure 4A). The first order of Ascomycota, Pezizales, ranks seventh, with only 2.05% of fungal records in FongiBase (Figure 4A). When considering the corresponding diversity of species within the previous orders, Agaricales ranks first (4423 taxa), followed by Helotiales (650 taxa), Polyporales (513 taxa) and Pezizales (498 taxa; Figure 4B).

In FongiBase, the three most frequently recorded ectomycorrhizal species are *Amanita rubescens* Pers. (6413 records, 0.6% of the total number of records), *Amanita citrina* Pers. (5716 records, 0.6% of the total number of records) and *Russula cyanoxantha* (Schaeff.) Fr. (5266 records, 0.5% of the total number of records; Appendix A). For plant saprotroph fungi, the highest number of observations concerns *Hypholoma fasciculare* (Huds.) P. Kumm. (7091 records, 0.7% of the total number of records), *Trametes versicolor* (L.) Lloyd (5915 records, 0.6% of the total number of records) and *Stereum hirsutum* (Willd.) Pers. (5495 records, 0.5% of the total number of records; Appendix A). The 21 first most recorded taxa belong to Basidiomycota (Appendix A). The first taxon in the Ascomycetes, *Xylaria hypoxylon* (L.) Grev., accounts for only 3461 records (0.3% of the total number of records (Appendix A).

### 3.6. Distribution Patterns of Fungal Species Occurrence

In FongiBase, the distribution of taxa occurrence shows a highly uneven pattern (Figure 5A). When considering the distribution of classes of abundance by multiples of hundreds of records, 88.2% of taxa (i.e., 11,252 taxa) are represented by less than 100 occurrences, and 94.3% of taxa (i.e., 12,021 taxa) account for less than 300 occurrences (Appendix A). At the other extreme of the distribution, only 196 taxa (i.e., 1.5%) are represented by more than 1000 occurrences (Appendix A).

When focusing on taxa with less than 100 occurrences, the distributions show again a highly uneven pattern. Thus, 7807 taxa (i.e., 61.2% of the total number of taxa) are represented by fewer than 10 occurrences (Appendix A). For taxa accounting for fewer than 10 records, the distribution consistently shows an unbalanced pattern, with respectively 2612 (i.e., 20.5% of the total number of taxa) and 5015 (i.e., 39.4% of the total number of taxa) taxa, with one or less than four occurrences (Appendix A).

When comparing this distribution of species abundance with nine theoretical models, we obtained the best fit with Pareto power law distribution (AIC = 96,731.4), followed by power Bend distribution (AIC = 100,500.7) and Fisher’s logseries distribution (AIC = 101,805.5; Figure 5B), suggesting on overrepresentation of rare taxa across the rank-occurrence distribution.

On average, French departments contain 24.7 ± 34.1 taxa represented by only one record along the two centuries of data acquisition (Appendix A). The detail of the geographic distribution of occurrence shows four clusters scattered across France, containing a high number of taxa with one record (Figure 6A). The first cluster consists of four administrative units located in the Atlantic ecoregion (the Western cluster, encompassing Loire-Atlantique, Maine-et-Loire, Vendée and Deux-Sèvres; Figure 6A and Appendix A). The second cluster, located in the same ecoregion, encompasses four administrative units (the Northern cluster, encompassing Pas-de-Calais, Nord, Somme and Oise; Figure 6A and Appendix A). The third Alpine cluster comprises six administrative units located in the Continental (i.e., Ain, Rhône and Loire) and Alpine (i.e., Haute-Savoie, Savoie and Isère) domains (Figure 6A and Appendix A). Last, four administrative units in the Continental ecoregion (the Eastern cluster, encompassing Doub, Jura, Haute-Saône and Territoire de Belfort) compose the last cluster. At the national level, the highest number of singletons (i.e., 199) is located in the department of Savoie (Figure 6A). The four previous clusters apart, six departments located in Continental (i.e., Haut-Rhin, Doubs, Côte-d’Or and Creuse) and Mediterranean (i.e., Hérault and Var) ecoregions harbor higher numbers of singleton (Figure 6A).

When considering the proportion of singletons per administrative units, French departments contain, on average, 0.4 ± 0.5% of records related to taxa observed only once (data not shown). The analysis reveals only one cluster with a high proportion of singletons, comprising three Mediterranean departments (i.e., Alpes-de-Hautes-Provence, Var and Bouches-du-Rhône; Figure 6B). In addition, five departments located in Atlantic (i.e., Val-d’Oise), Continental (i.e., Ardenne, Côte-d’Or and Creuse) and Mediterranean (i.e., Corse-du-Sud) ecoregions show a high proportion of taxa represented by less than one occurrence (Figure 6B). At the national level, the highest proportion of singletons (i.e., 2.5) is located in the department of Creuse (Figure 6B).

At the national scale, the number of observations per department is positively correlated with the number of singletons per department (*r* = 0.68; *p* < 0.001 by Spearman correlation test; Figure 7A), and negatively correlated with the proportion of observations represented by singletons per department (r = −0.32; *p* = 0.001 by Spearman correlation test; Figure 7B).

### 3.7. Ecology of Fungal Records

In all, 917,641 records (i.e., 87.9%) could be unambiguously ascribed to an ecological guild sensu FunGuild, and 12,2865 records (i.e., 11.8%) could be satisfactory assigned to a habitat of collecting sensu CORINE biotopes codification using the botanical information provided by the observers at collecting time. When combining the two descriptors, 110,106 records (i.e., 10.5%) could be characterized by both their biotope and their belonging to one of the 12 fungal ecological guilds of FunGuild (Figure 8A).

Among the functional diversity of fungi, three ecological guilds are overrepresented in FongiBase, namely Mycorrhizal (409,643 records, 44.6% of records assigned in FongiBase), Plant saprotrophs (289,921 records, 31.6%) and Wood saprotrophs (166,402 records, 18.1%; Figure 8B and Appendix A). Contrastingly, nine remaining ecological guilds account for only 5.7% of the records assigned in FongiBase (Appendix A). For the 12 fungal guilds, records from forests are the most represented, with 71.3% ± 23.8% of the total number of records (from 29.9% for Bryotrophic to 100 % for Animal saprotrophs and epiphytes; Figure 8A).

Among habitats represented by fungal records, Forests are overrepresented with 93,698 (76.3%) observations (Figure 8B and Appendix A). The most sampled types are Mixed woodlands, Broad-leaved deciduous forests and Coniferous woodlands, with 44,525 (36.2%), 25,289 (20.6%) and 18,008 (14.6%) records, respectively (Figure 8B and Appendix A). Contrastingly, Broad-leaved evergreen woodland account for only 0.1% of the total number of records (179 records; Appendix A).

Within the distribution of frequency, Urban parks and large gardens comes fourth, with 5335 (4.3%) records. The type Fens, transition mires and springs comes after, with 4877 (4%) records, followed by Dry calcareous grasslands and steppes, with 3687 (3.0%) records, Alluvial and very wet forests and brush, with 3543 (2.9%) records, Alpine and subalpine grasslands, with 3406 (2.8%) records, and Coastal sand-dunes and sand beaches, with 2474 (2.0%) records (Figure 8B and Appendix A).

### 3.8. Distribution Patterns of Contributors

In all, FongiBase contains 26 distinct contributors (Appendix A). The two highest numbers of records originate from FMBDS (Fédération Mycologique et Botanique Dauphiné-Savoie) and CBNFC-ORI (Conservatoire Botanique de Franche-Comté Observatoire Régional des Invertébrés), with 180,364 (17.3%) and 177,839 (17%) records, respectively (Appendix A and Figure 8). Four other channels of fungal observations account for more than 10% each, namely MNHN (Museum National d’Histoire Naturelle), RNF (Réserves Naturelles de France), ADONIF (Association pour le Développement d’Outils Naturalistes et Informatiques pour la Fonge and SMNF (Société Mycologique du Nord de la France) who respectively produced 159,878 (15.3%), 138,521 (13.3%), 121,231 (11.6%) and 112,922 (10.8%) records (Appendix A and Figure 9). The 20 remaining contributors produced each less than 5% of data registered in FongiBase (Appendix A).

When considering the relationship between the number of records in FongiBase and the number of taxa with one observation (singleton taxa), the obtained pattern opposes three highly active structures who show an underrepresentation of singletons (i.e., CBNFC-ORI, RNF and MNHN, with 0.02, 0.06 and 0.1% of singletons, respectively) to three modestly contributing structures who present an overrepresentation of singletons (i.e., SMF, Société Mycologique de la Roche sur Yon-SMRY and Asco France with 1.2, 2.3 and 10% of singletons, respectively; Appendix A and Figure 9). Located at an intermediate position, four structures (i.e., FMBDS, ADONIF, SMNF and AMO-UL) are characterized by a moderate singleton ratio (Appendix A and Figure 9).

## 4. Discussion

Our report analyses the spatio-temporal, taxonomic and ecological patterns underlying the remarkable species richness of fungi inventoried in the French national mycological database. The analysis revealed four main facets of the distribution of more than one million fungal records gathered across the French metropolitan territory since the 19th century.

First, the chronology of fungal data acquisition since the early 19th locates the late 1970s as a breaking point separating (1) ancient data of erratic distribution with low usability for conservation, and (2) modern records associated with high sampling effort at local scale. Second, at the national scale, the geography of fungal records is highly uneven, with four main clusters of intense data acquisition contrasting with the poorly documented southern part of France, including the Mediterranean domain. Third, the distribution of taxa occurrence shows an overrepresentation of poorly observed/observable organisms, with a majority of taxa recorded less than 10 times. Last, from an ecological perspective, the dataset shows a highly unbalanced distribution of observations, with more than three-quarters of records from forest ecosystems, mainly in the ectomycorrhizal guild, while only 4% of data document fungal patterns in grasslands.

Considered altogether, these characteristics of the French fungal database highlight the strength of the analyzed dataset, but also alert us about substantial knowledge gaps and methodological issues that should be considered in future strategies aiming at promoting the conservation of fungal diversity in France.

### 4.1. French Metropolitan Territory, a Species Rich Area for Fungi or a Mycologist-Rich Area?

As primarily assumed, the French metropolitan territory experiences a particularly high diversity of fungi as compared to most European countries, with more than 7000 expected taxa out of 15,000 European ones [55]. We here confirm this 15-year-old estimation, by attesting the presence of 11,556 taxa in the French National checklist out of 1,043,262 observations. The analyzed territory a priori appears as remarkably rich, as compared to other European countries of similar sampling effort (i.e., measured as the number of fungal records), including the United Kingdom (6453 taxa), Germany (5531 taxa) and Switzerland (4974 taxa; [7]). This difference in specific richness is unlikely to be explained by differences in sampling efforts among European countries, as reflected by the absence of correlation between the number of taxa and the number of records in France and in the nine countries included in FunClim European database (R = 0.46 & *p* = 0.22; Figure 1). Explicitly, the French database harbors a particularly high density of taxa per record, with 1.11 taxa per 100 records, when 100 fungal observations account for 0.46 and 0.42 taxa in the United Kingdom and Germany, respectively (Figure 1).

More likely, the high fungal specific richness recorded in FongiBase may reflect the large area of the French metropolitan territory as compared to neighboring inventoried countries, and the gamma diversity associated with its remarkable diversity of represented ecoregions (Appendix A) and natural habitats (Figure 8A). In our analysis, we found a positive and significant correlation between the country area and specific richness of the 10 analyzed territories (R = 0.79 & *p* = 0.006; Figure 1). Interestingly, this correlation remains highly significant (R = 0.98 & *p* = 2.1 × 10^−8^), when considering data from the United States (i.e., 44,488 taxa in 24,930,000 km^2^ from 2,200,000 records; [8]).

This result may also traduce the existence of a particularly efficient network of mycological associations developed (as compared to neighboring countries [55]) who share a taxonomically vast mycological expertise across a wide range of fungal lineages. Our analysis does not support this hypothesis, based on the results of a GLM procedure (Table 2). However, this absence of correlation between the density of the network of mycologists and the taxonomic knowledge recorded in the corresponding area may be considered with caution because of the intrinsic limits of both the database and the used method of analysis. Thus, 77 out of the 92 metropolitan departments do not accommodate any established contributor, while they all harbor fungal records in FongiBase. This pattern may first reflect the ability of field mycologists to realize long-distance surveys, to collect data in territories of high mycological interest (see [16] for Montpellier area).

At the department level, our analysis revealed the existence of four clusters showing a high intensity of fungal data recording during the last two centuries (Figure 2). Altogether, these clusters account for 66.62% of the total number of located records at the department scale (Appendix A), while they represent only 17.12% of the metropolitan French Territory. Similar uneven geographic densities of fungal records have been previously observed across Europe. Using 7.3 million national-scale fruit body records from nine countries, ref. [7] interpreted this consistent pattern as a consequence of variation in sampling intensity. The determinism is complex and may include sociological as well as ecological determinants. Based on citizen science contribution to large dataset, similar unbalanced recording density was previously reported for Fungi from Denmark [6]. In a dataset assembling 292,022 records, these authors reported a negative correlation between human population density and mycological data density. In Fongibase, the network of mycological societies (i.e., ADONIF, AMO, FMBDS, SMNF, ABMARS and CBNFC-ORI) provided the totality of records in three of the four clusters of mycological data (Figure 9 and Appendix A). Contrastingly to amateur mycologists, academic institutes such has the French Museum of National History and ADONIF highly contributes to the dataset (Figure 9 and Appendix A) but based on a highly diffuse way across the territory, by providing mycological data in all the 96 French departments.

Altogether, these results suggest that the patterns of mycological forays more than the location of their mycological societies shape the geography of data recording in France. This analysis also highlights the complementarity between the multiple contributors of FongiBase, including museums, educational institutions, mycological societies, and citizens to nourishing the National database. Complementary research, including sociological data and composite network analyses [56], is needed to provide a more accurate view of the links between mycologist recording activity, fungal diversity and landscape characteristics.

### 4.2. Red List Initiatives Drive Fungal Recording Activity

In the four clusters of fungal recording, most data were acquired since 1977, but with slight differences among clusters (Appendix A). In three of them, a decline in field observation currently follows a maximum of sampling intensity recorded in 1997 (11,143 records), 2010 (14,267 records), 2014 (8130 records) in the northern cluster, the eastern cluster and the western cluster, respectively (Appendix A). Contrastingly, the Alpine cluster shows a positive dynamics of field data recording, with a maximum of 11,566 data registered in 2020.

In the three clusters showing a distinct maximum of recording activity (i.e., northern, eastern and western clusters), with a more saturated pattern of the taxa accumulation curve (Appendix A), the observed temporal dynamics of fungal data recording corresponds to the emergence of three regional red lists. Thus, the Nord-Pas-de-Calais, Franche-Comté and Poitou-Charentes regional red lists were published in 1997, 2013 and 2018, respectively (Figure 3B). These data suggest that these conservation initiatives, supported by local mycological societies, drove an intensified dynamics of field observation and systematic archiving of mycological data before the publication. In these three areas, a marked decline of data recording was observed after red list publication, suggesting a transient boosting effect of conservation initiatives on field data archiving, and a need of periodic updating of regional red lists to maintain recording activity. Interestingly, the alpine cluster shows a contrasted pattern of both temporal dynamics of data recording (absence of recording pick) and state of advancement of regional conservation initiative (absence of red list; Appendix A).

### 4.3. A Two-Century Long Dynamic of Fungal Data Recording

Our analysis of the temporal dynamics of FongiBase revealed a four phase-pattern, with the major part of fungal observation being recorded since 1977 (Figure 3A,C).

The most ancient period extends from 1800 to 1886, and corresponds to a highly erratic pattern of fungal recording (Figure 3C), with few records per year (41.8 records per year), resulting in a modest contribution to the current database (0.34% of the total number of records, for 3522 data). These records correspond to early contributions of particularly active mycologists (e.g., L. Graves, which provided a significant contribution in 1857 in the Picardie department; Appendix A).

These historical records, and their related herbarium materials, are of primary importance for science as valuable resource for a wide variety of requests [57]. First, they constitute high-value biological samples for taxonomic updating using next-generation sequencing tools (e.g., [58] for clarification of *Cortinarius* subgenus *Leprocybe* in Europe). They also provide deep temporal perspective for investigating the effect of climate deregulation on biodiversity patterns (e.g., [5,16,17]; see also [18] for a review). However, ancient data have to be considered with caution, as their quality (accuracy of records, taxonomic identification, etc.) is highly dependent on the context of the sampling (e.g., state of knowledge, quality of material archiving [59]; [19]). From a mycological history perspective, 19th century fungal observations were registered by an unstructured constellation of isolated mycologists–botanists scattered across the France territory (e.g., de Candolle, Raffeneau-Delile and Dunal in Montpellier area [16,37]; [60] in Northern France; Figure 3C). Thus, the end of the 19th century corresponds to the emergence of ambitious initiatives to assemble talented naturalists into mycological societies, at both the national (creation of the French Mycological Society in 1884) and the regional scales (e.g., creation of the Société d’Horticulture et d’Histoire Naturelle de l’Hérault in 1860). This dynamic probably drove the observed increasing in field data collecting and archiving, as suggested by the diversification of the fungal data sources observed in Fongibase (Figure 3C).

The second period extends from 1887 to 1977, and accounts for 3.6% of the total number of records (Figure 3C). The observed recording pattern during this second phase is also highly erratic, but with a 10-fold higher recording activity (422 fungal records per year) as compared to the previous phase. During this period, fungal recording activity was fed by an unachieved network of mycological societies (Figure 3C), with a main contribution of the Société Mycologique de France (SMF), the Fédération Mycologique et Botanique du Dauphiné-Savoie (FMBDS, created in 1960) the Association Mycologique de l’Ouest (AMO, created in 1952) and the Société Mycologique du Nord de la France (SMNF, created in 1967; Figure 3C).

From 1977 to 2007, a marked increasing of data recording emerged from the analysis, with 44.4% of the total number of observed archived in Fongibase (Figure 3C), corresponding to a recording intensity 37-fold higher (15,441.7 records per year) as compared to the first part of the 20th century (i.e., the second phase). This period coincides with the structuration of an extensive network of mycological associations, concomitantly with the progressive emergence of botanical conservatories as complementary contributors of FongiBase. Interestingly, a similar temporal dynamic was reported by [7], who highlighted a high concentration of records since 1960 in the 6 million data assembled from Germany, the U.K., Denmark, Norway, Austria, Switzerland, Estonia, Slovenia and the Netherlands observation networks. This part of FongiBase constitutes a highly valuable dataset for investigating mid-term change in macrofungal patterns at the scale of France. In particular, this corpus may be mobilized in order to feed red list initiatives as well as European global conservation initiatives [61], and large-scale assessments of the effect of global change on European fungal diversity [18].

A majority (i.e., 51.6%) of fungal records assembled in Fongibase have been acquired since 2007 (Figure 3C). In contrast with the period 1977–2007, the last 15 years did not show a progressive increasing in recording activity, but rather a stabilization followed by a sharp drop since the 2016 recording maximum (60,275 and 33,178 records in 2016 and 2018, respectively; Figure 3C). The determinism of this recent decline (at least in 2017 and 2018, i.e., before the beginning of the COVID-19 pandemic) remains unclear, and the short duration of this pattern urges caution about its possible fitting in a long-term perspective. However, this trend should be considered with attention, as a similar trend was observed in most of the European countries analyzed by [7]. The stabilization observed since 2007 may reflect (1) biological processes related to a decline of the intensity of fruiting events across the analyzed territory in a context of increased drought [12,62,63], and/or (2) the sociology and the anthropology of the network of French amateur mycologists, who faces the deleterious consequences of specialist aging and a chronic deficit of young promising recruited, but also (3) a decreased activity of the main contributors following red list initiatives (Appendix A), whose implication in the long-term may be sustained by academic channels and official acknowledgement. These three scenarios may act simultaneously, and their respective contribution to the observed pattern remains to be evaluated.

### 4.4. A Deficit of Fungal Record in the French Mediterranean

In this analysis, we identified a contrasted recording effort depending on the four ecoregions represented across metropolitan France, with a lower number and density of records per department in the Mediterranean as compared to Continental, Atlantic and Montane domains (Table 1). Interestingly, the Mediterranean shows a similar species richness as compared to other domains, but with a substantially lower sampling, ranging from 7.1% and 22.3% of the number of records gathered in the Atlantic and the Alpine domains, respectively (Table 1). Considered altogether, these results depict (1) the unbalanced sampling effort across French ecoregions, (2) the Mediterranean as a poorly represented area in the French fungal database and based on this deficit of records, as the potentially richest area for fungal diversity in France.

### 4.5. Beyond the Data: An Unbalanced Taxonomy of Records

Our analysis showed a highly even distribution of fungal lineages, with an overrepresentation (86.8%) of records belonging to Basidiomycota (Figure 4). This result reflects the well-documented dominance of Basidiomycota lineage in the fruiting patterns in temperate ecosystems, as consistently reported in the literature of diachronic surveys in the northern hemisphere (e.g., [21,26,64,65,66]).

At the lower taxonomic level, Agaricales, Russulales, Polyporales and Boletales account for 77.1% of the records in FongiBase (Figure 4A), while the two most observed orders of Ascomycetes (i.e., Pezizales and Helotiales) account only for 3.9% of records. Contrastingly, the corresponding diversity of species shows a more balanced pattern, with two orders of Ascomycota (i.e., Helotiales and Pezizales; ranks 2 and 4) intercalated between Agaricales (rank 1) and Polyporales (rank 3; Figure 4B).

This pattern appears highly similar with data that were previously reported by [7] who observed the same sequence of orders dominating the recorded taxonomic diversity in Central-Western European countries. In particular, [7] estimated between 49 and 57% the proportion of Agaricales records in Central-Western European countries, followed by Russulales and Polyporales (Estonia apart, where Pezizales rank two after Agaricales). The respective biological (distribution of fungal taxa) and sociological (skills and “palatability” of field mycologists) determinants of this highly consistent pattern across Europe may be explored to fully understand and evaluate fungal conservation stakes at global scale.

### 4.6. Beyond the Data: An Unbalanced Ecology of Records

In Fongibase, our analysis revealed an unbalanced pattern of the representation of fungal guilds, with a marked dominance of mycorrhizal species in the database (Figure 8A). This pattern reflects (1) the belonging of a majority of Agaricales, Russulales and Boletales to the ectomycorrhizal guild [67] and (2) the central place of forests dominated by ectomycorrhizal trees (i.e., Fagaceae, Pinaceae, Betulaceae and Salicaceae) in the fields which are classically visited by French recorders (Figure 8A), and more generally by mycologists during forays [6].

This discrepancy among mycorrhizal and other fungal guilds was not confirmed at the species level. Thus, among the 10 most observed species, five (i.e., *Hypholoma fasciculare*, *Trametes versicolor*, *Stereum hirsutum*, *Pluteus cervinus* and *Xylaria hypoxylon*) are wood decayers, while only four (i.e., *Amanita rubescens, Amanita citrina*, *Russula cyanoxantha* and *Laccaria amethystina*) are mycorrhizal. Interestingly, the two most frequent species, namely *Hypholoma fasciculare* and *Amanita rubescens*, were also recorded as widespread and abundant across the Central-Western European countries by [7]. Our analysis confirms their status as potentially efficient fungal markers of climate change in this part of the world.

In FongiBase, the ecological context of fungal recording was poorly documented, with only 11.8% of records which could be assigned to a natural habitat *sensu* CORINE Biotopes. When considering the 122,865 records which could be satisfactory assigned to a natural habitat, our analysis showed a strong deficit of observations in (1) open vegetations with high conservation stakes (10.1% of assigned records), such as dry calcareous grasslands (3% of records) and in (2) artificial habitats (6.4% of assigned records) with high sociological stakes, such as urban green spaces (4.3% of records), cultivated landscapes (0.1% of records) and cities/villages (0.01%; Figure 8B). The observed pattern may be explained by both a low level of interest in these habitats by amateur mycologists and by the presence of particularly poor communities of macrofungal species in these artificialized systems.

The observed lack of ecological metadata associated with fungal records may be explained by the absence of a standardized procedure of fungal data recording, which may include a systematic characterization of the botanical context of the record. This characteristic of FongiBase limits its efficiency to provide a comprehensive overview of the real sampling effort devoted to mycological knowledge across the territory. However, the under sampling of agricultural landscapes for its fungal diversity was previously recorded in Denmark [6]. In 45% of the France area devoted to agriculture [68], only 144 fungal records are currently archived, making this part of the landscape “the great absentee” of the base.

### 4.7. Between Fungal Rarity and Sampling Activity (Methodological Development)

In Fongibase, a majority of species (61.2%) are represented by fewer than 10 records, and 20.5% by only 1 observation, resulting in a Pareto distribution as the best fit theoretical model (Figure 5). This dominance of rarely observed species in fungal communities has been widely observed at the local scale based on either mycelium in soil (e.g., [25,26]), leaf tissues [27] and fruit bodies (e.g., [69,70]). The observed Pareto distribution may reflect an under sampling of the fungal diversity within the corresponding territory. Thus, in a study based on citizen science records in England and in Switzerland, [31] reported a lognormal distribution of records at the national scale, contrasting with a Pareto law distribution for local scale data. This author suggests that the observed lognormal distribution of large-scale data may be biased by the used citizen science approach, which favors repeated observations of easily identifiable species [31]. Our analysis does not support this last hypothesis, as the FongiBase subdataset compiling fungal records from citizen science (i.e., records from CBNFC-ORI, MNHN and RNF) fits a Pareto law distribution.

Remarkably, our analysis shows that the number of singletons observed across the territory increases as the sampling effort increases (Figure 7B), while the proportion of singletons concomitantly decreases (Figure 7A). This relationship is significant, and follows a logarithmic model (Figure 7A,B). This result suggests that the number of fungal species observed only once may be a good proxy of the quality of sampling effort, but the proportion of these rare species may inversely indicate the limit of knowledge of fungal communities at local scale. This pattern may also suggest that the dominance of rarely observed species may be not artefactual, but in relation with the inherent characteristics of fungal communities at local scale. As a practical consequence of the above relationship between sampling effort and singleton patterns, the four clusters of fungal recording determined by our analysis appear as (1) the most efficient providers of rare species (Figure 6A), as a consequence of the activity of a concentrated network of local associations (Figure 6A) and (2) the regions where the proportion of rare species in the dataset is the lowest (Figure 6B).

When considering the wide range of contributors of FongiBase (Figure 9), a contrast between three types of contribution arises, based on the characteristics of the generated subdataset. A first cluster of specialized mycologists, containing AscoFrance, SMF and SMRY, shows an ability to generate high-definition records in small datasets with many rare species (Figure 9). At the other extreme of the range, a group of highly active contributors, deploying citizen science methods across the territory, and containing CBNFC-ORI, RNF and MNHN, shows an ability to provide large datasets of frequent species (Figure 9). Located at an intermediate position between these two groups (Figure 9), a cluster of generalist mycologists, containing ADONIF, FMBDS, SMNF, AMO and UL, stands out by its ability to provide large dataset containing a high number of rare species, based on an intense sampling activity. These three types of fungal diversity observers, and the richness of the corpus they contributed to assemble in FongiBase, reflect the diversity of the French community of mycologists, and their complementary to synergistically feed large-scale initiatives.

## 5. Conclusions

The heterogeneous nature of the dataset assembled by the variety of contributors of FongiBase, considered with the heterogeneous geography and ecology of data at both departmental and ecoregional scales, brings to light two conclusions. First, highly structured networks of mycological associations played a pivotal role in the French national mycological dataset by providing high-quality fungal records recoverable in the first regional red list initiatives. Second, there is a need to sustain the development of such organizations and help stimulate fungal recording activity in poorly documented areas, including the Mediterranean ecoregion and orphan habitats, such as agricultural landscapes, urban areas and open vegetation preceding forest establishment.

## Figures and Tables

**Figure 1 jof-08-00926-f001:**
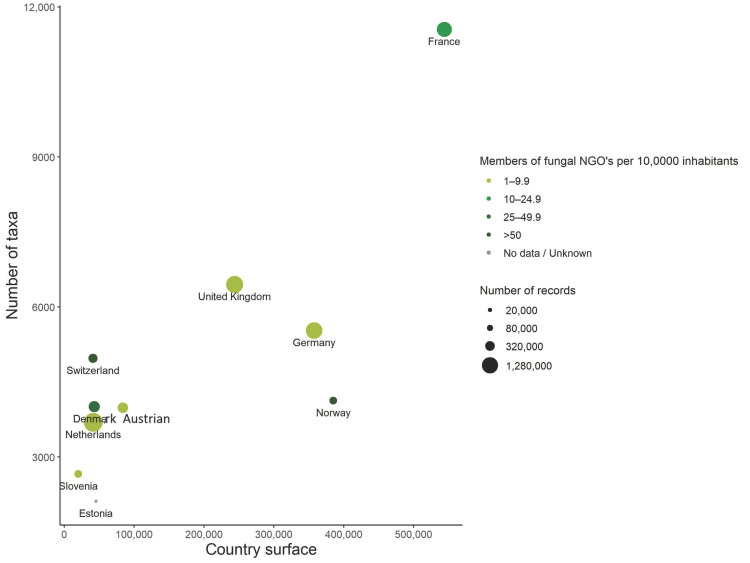
Distribution of national databases introduced in [7] and FongiFrance national database, in relation to the number of taxa and the country surfaces with the number of records and the number of mycological societies per 100,000 inhabitants [16].

**Figure 2 jof-08-00926-f002:**
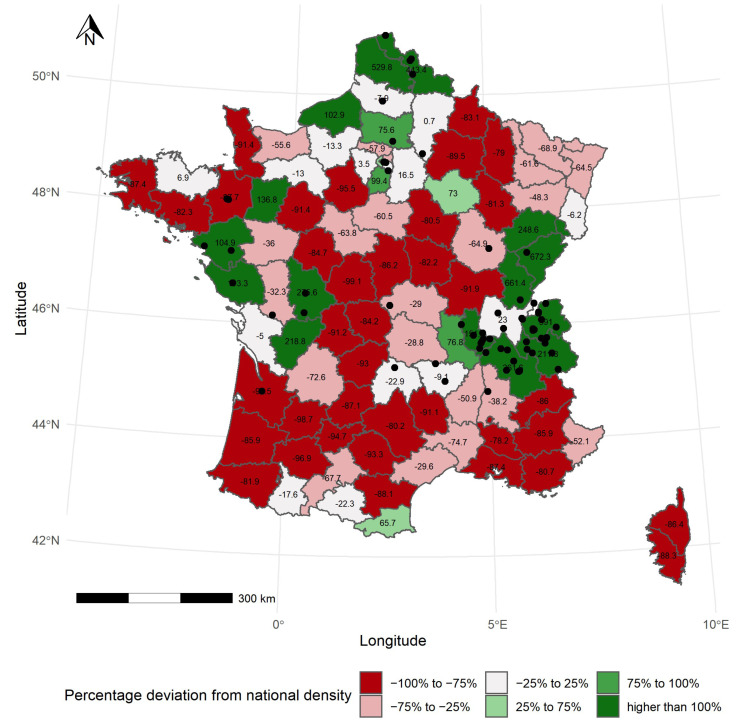
Distribution of fungal records across administrative units in France. The gradual color variation represents the difference between the density of recorded observations of each administrative unit and the density at the national level, expressed in percent. The highly localized administrative units of Ile-de-France are not represented. For these departments, values are respectively 318.54 (Paris), 51.71 (Hauts-de-Seine), −49.69 (Seine-Saint-Denis) and −20.61 (Val-de-Marne).

**Figure 3 jof-08-00926-f003:**
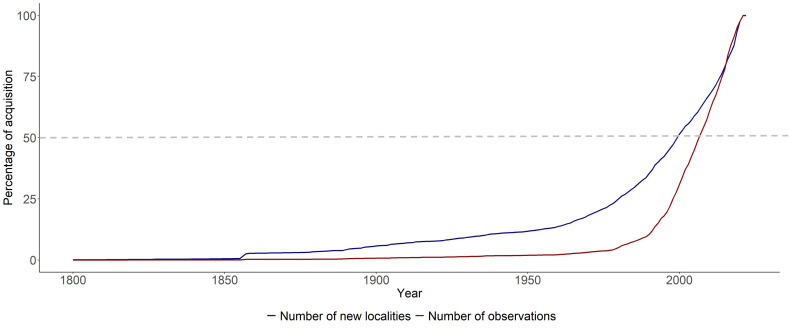
Temporal dynamics of fungal recording in France during the period [1800–2022]. (**A**) Accumulation curve of sampled sites and records during the whole period, expressed in percent. (**B**) Detailed representation of the period [1990–2022], with the position of main fungal conservation initiatives at global and national scales. (**C**) Temporal dynamics of recording per site during the whole period, with corresponding map of the distribution of fungal records across administrative units in France. Shaded areas represent the four steps of data acquisition dynamics, based on graphic analysis. Black dots indicate the location of mycological societies.

**Figure 4 jof-08-00926-f004:**
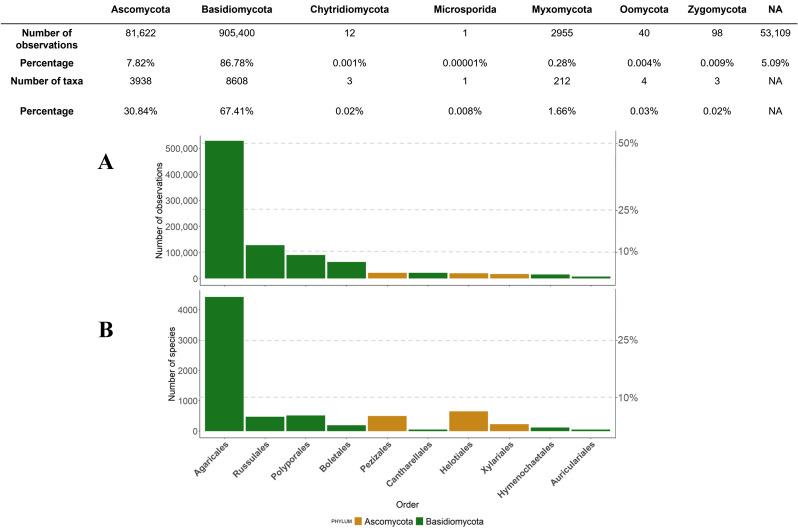
Distribution of fungal records in the database in relation with their taxonomic position at phylum and order levels. Color of bars indicates orders in the Basidiomycota (in green) or Ascomycota (in orange).

**Figure 5 jof-08-00926-f005:**
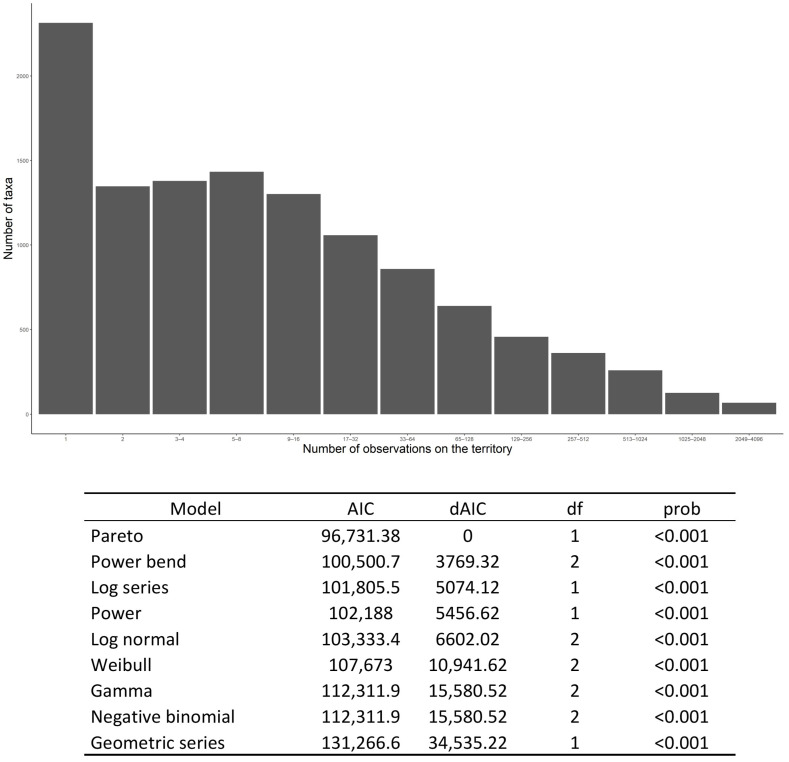
(**A**) Species distribution abundance (SAD) of fungal taxa in FongiBase and (**B**) Comparison of the distribution of taxa occurrence in FongiBase and nine theoretical distribution models of abundance. AIC: Akaike information criterion. dAIC: difference in AIC score between the best model and the model being compared. Df: degree of freedom. *P*: similarity between the observed distribution of data and the theoretical distribution of data for the corresponding model.

**Figure 6 jof-08-00926-f006:**
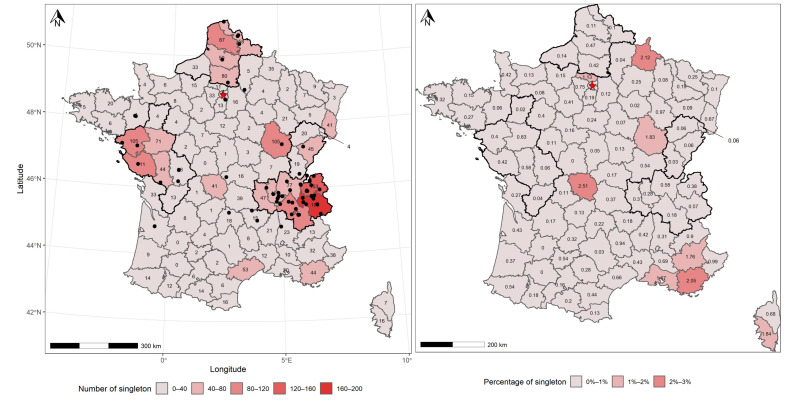
Spatial distribution of taxa represented by one record (singletons) in FongiBase, across French administrative units (departments) using (**A**) the number of singletons and (**B**) the proportion of singletons on the total number of records of each department, expressed in percent. Black dots indicate the location of mycological societies. The red star corresponds to the departments of Paris, Hauts-de-Seine, Seine-Saint-Denis and Val-de-Marne, which respectively contain 4, 2, 1, and 2 singletons, corresponding to a proportion of 0.038%, 0.011%, 0.004%, and 0.008% of records in each department.

**Figure 7 jof-08-00926-f007:**
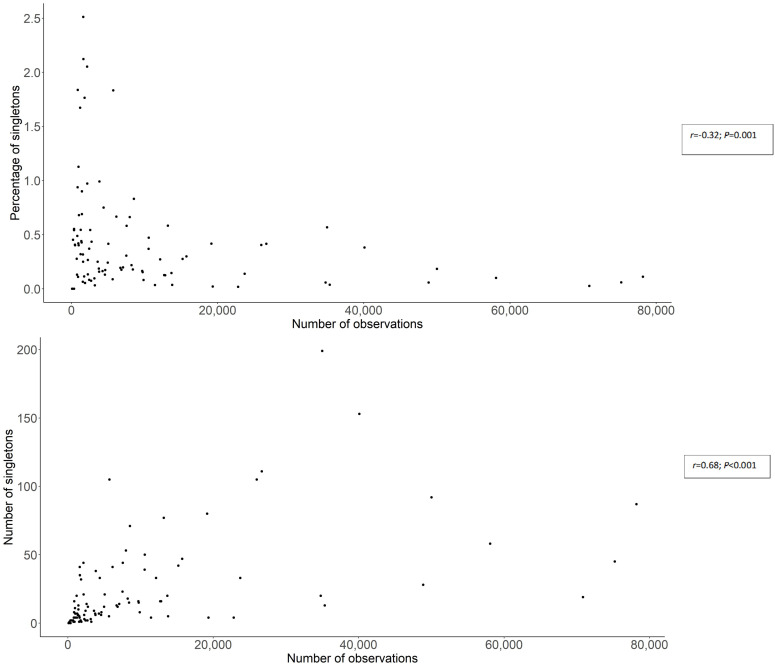
Correlation between the number of records in FongiBase in the 96 French departments and (**A**) the proportion of taxa represented by one record (singletons) and (**B**) the number of singletons, with the significance of Spearman correlation tests.

**Figure 8 jof-08-00926-f008:**
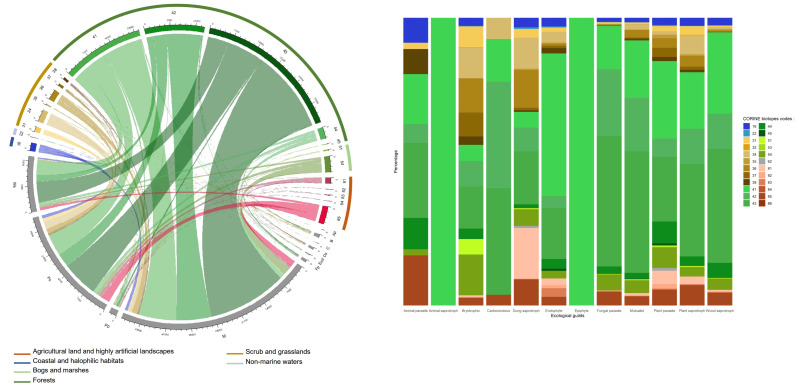
(**A**) Distribution of the functional diversity of fungal taxa recorded in FongiBase and the habitats of collecting. The typology of ecological guilds is based on FunGuild and includes 12 concatenated types: Animal parasites (Ap), Animal saprotrophs (As), Bryotrophic (B), Carbonicolous (C), Dung saprotrophs (Ds) Endophytes (End), Epiphytes (X), Fungal parasites (Fp), Mycorrhizal (M), Plant parasites (Pp), Plant saprotrophs (Ps) and Wood saprotrophs (Ws). The typology of habitats is based on CORINE biotope codification. All records which could be unambiguously assigned with confidence to one of the following CORINE codes are represented. 16: Coastal sand-dunes and sand beaches, 22: Standing fresh water, 31: Heath and scrub, 32: Sclerophyllous scrub, 34: Dry calcareous grasslands and steppes, 35: Dry siliceous grasslands, 36: Alpine and subalpine grasslands, 37: Humid grasslands and tall herb communities, 41: Broad-leaved deciduous forests, 42: Coniferous woodland, 43: Mixed woodland, 44: Alluvial and very wet forests and brush, 45: Broad-leaved evergreen woodland, 51: Raised bogs, 53: Water-fringe vegetation, 54: Fens, transition mires and springs, 62: Inland cliffs and exposed rocks, 81: Improved grasslands, 82: Crops, 83: Orchards, groves and tree plantations, 84: Tree lines, hedges, small woods, bocage, parkland dehesas, 85: Urban parks and large gardens, 86: Towns, villages, industrial sites. (**B**) Relationship between the fungal ecological guilds and CORINE biotopes habitats.

**Figure 9 jof-08-00926-f009:**
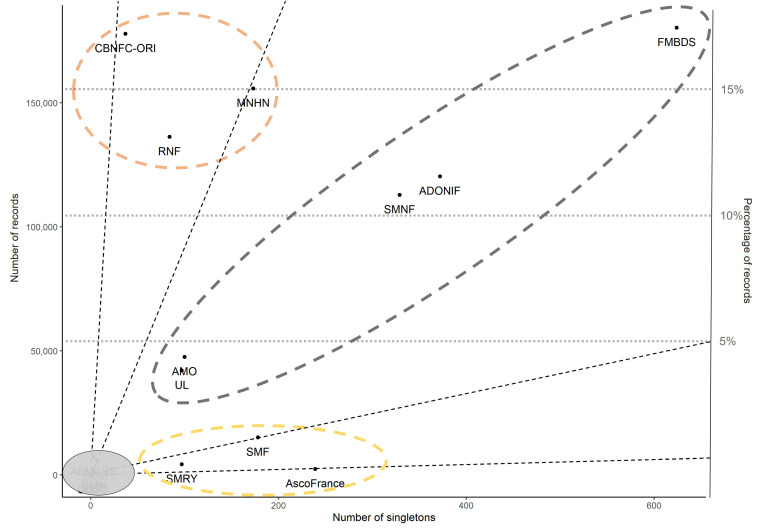
Distribution of the contributors registered in FongiFrance in relation to their corresponding numbers of records and taxa with one occurrence, including the representation of the whole panel of contributors. The area in grey represents a set of six contributors we provided less than ten thousand records each. Dotted lines correspond to singleton ratios of 0.001, 0.001, 0.01 and 0.1, from the bottom to the top of the panel.

**Table 1 jof-08-00926-t001:** Patterns of fungal records across the Atlantic, Continental, Mediterranean and Montane French ecoregions. Letters in bold refer to results of Tukey pairwise comparisons, with different letters indicating significant differences between ecoregions (*p* < 0.05). The rarefied species richness refers to the number of species recorded for a normalized number of 33,156 records in the four different ecoregions (corresponding to the number of observations identified at the species level in the Mediterranean ecoregion).

Parameter	Ecoregion				
	Atlantic	Continental	Mediterranean	Montane	Total France
Number of observations (n)					
Total	492,014	342,124	35,202	157,558	1,043,262
Mean per department	10,696 **a**	11,404 **a**	3200 **b**	17,506 **a**	
Standard deviation	16,343.08	18,262.98	2594.12	18,861.16	
Observation density (n·km^−2^)					
Mean per department	2.06 **a**	2.72 **a**	0.58 **b**	3.18 **a**	
Standard deviation	2.68	4.71	0.41	3.28	
Specific richness (S)					
Mean per department	1274.2 **a**	1400.03 **a**	1045.46 **a**	2099.56 **a**	12,055
Standard deviation	941.32	808.49	503	1605.13	
Rarefied species richness	3632.17	3686.14	4400	478.96	
Sampling effort					
Number of communes	16,815	12,258	2771	2992	34,836
Number of sampled communes	5147	3912	1081	1396	11,536
% of sampled communes	30.6	31.9	39	46.7	33.1

**Table 2 jof-08-00926-t002:** Analysis of the correlation between the number (a) and density (b) of fungal species per km^2^ at the department scale and the number and density of records per km^2^, and number and density of fungal species per km^2^, the number and density of inhabitants per km^2^ and the number of contributors in FongiBase, using a generalized linear model procedure. The analysis excludes four departments of the region Ile de France which are totally included in Paris urban area (i.e., Paris—75, Hauts-de-Seine—92, Seine-Saint-Denis—93, and Val-de-Marne—94) from the rest of the territory.

Coefficient	Estimate	Standard Error	z Value	*p*-Value
(a)				
Intercept	−4.04	2.97	−1.36	0.17
Nb of records (log)	**1.20**	**0.35**	**3.39**	**0.0007 *****
Nb of inhabitants (log)	**0.53**	**0.23**	**2.29**	**0.02 ***
Nb of contributors	0.0344	0.02	1.56	0.12
Nb of records (log) x nb of inhabitants (log)	**−0.05**	**0.03**	**−1.98**	**0.04 ***
(b)				
Intercept	−2.60	1.13	−2.30	0.02 *
Record density (log)	**0.50**	**0.16**	**3.12**	**0.0018 ****
Density of inhabitants (log)	0.25	0.24	1.05	0.29
Nb of contributors	−0.01	0.08	−0.09	0.93

Bold characters indicant significant correlations (*: *p*-value < 0.05; **: *p*-value < 0.01; ***: *p*-value < 0.0001).

## Data Availability

All data analyzed in the study are available online using https://fongifrance.fr/fongi2022/ (accessed on 22 July 2022).

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
