# Peer review of "Unravelling the French National Fungal Database: Geography, Temporality, Taxonomy and Ecology of the Recorded Diversity"

_jof, 2022, doi:10.3390/jof8090926_

Round 1
Reviewer 1 Report
This paper deals with a large series of data on fungal records in France from mainly the 19th century to the present day. The reading of the manuscript shows the exhaustive analysis of the data as well as their evaluation and discussion. The methodology used was already proposed by the authors in previous experiments and articles and seems adequate and reproducible for the purpose of the study. However, there are some shortcomings in the formal presentation of these results in the form of figures and tables that should be remedied by the authors to affect the quality of the article. Finally, some suggestions and comments on the results obtained and their discussion should be noted.
The research presented is in the scope of Journal of Fungi and therefore publishable. However corrections are recommended before being accepted for publication.
Globally the methodology and bibliography seem accurate and suitable some minor corrections are needed in some figures and tables, and in the formatting of some parts of the text (e.g., font size, scientific names).
Specific comments for the authors:
a) Abstract
Page 1, line 22. The percentage attributed to Agaricales (39.7%) does not correspond to that attributed to it (50.4%) in the Results section (page 9, line 307). The authors should clarify this discrepancy.
b) Materials and methods
Page 3, lines 130-133. It would be useful to indicate the table number to clarify which table is referred to in the text. The "second table" named in the text does not correspond to the information shown in Table 2 or Table S2.
Page 4, lines 146-148. Authors should pay attention to habitat assignment of records according to georeferencing and overlap with CORINE biotopes using geographic information systems may be incorrect. This could be even more important if we consider that Mediterranean environments may be more prone to changes in these biotopes (e.g. those resulting from fires).
Page 4, lines 174-175, 178-180. (Also in the "Discussion" section, page 17, line 588). It seems that the font size is larger than the rest.
Pages 4, line 177. About generalized linear model, have the authors considered that the number and density of fungal species might not follow a normal or parametric distribution? In assessing the relationship between the percentage / number of singletons and the number of observations, the authors do assume that these data follow a non-parametric distribution.
c) Results
Pages 4, line 188. In the 11556 taxa, were the synonyms of the scientific names considered and screened according to Fongibase? If so, the authors should indicate this in the text.
Page 10, lines 317-326. As this is the first time that scientific names appear in the text, the should bear the abbreviated name of the naming author(s).
Page 10, line 322-323. According to Table S4, what about Stereum hirsutum?
Page 13, line 395. What do the authors mean by the term "satisfactory"? If they refer to records with precise GPS coordinates, they should indicate this in the text.
Pages 14 and 15, lines 449-457. The authors could comment on the case of the ULCO organisation, with only 33 registrations and 2 singletons, representing 6.1% of the singletons, according to Table S1.
d) Discussion
Page 15, lines 476-479. Is the dataset used to analyse the habitat type of each of the records, as well as their life type, data with accurate GPS coordinates? In the material and methods section (page 3, lines 125-127) it is stated that less than half of the records were taken with precise geographical coordinates. Perhaps this bias in the data used to assess habitat may have influenced this overrepresentation of forest environments. Or perhaps the authors consider that there is no such bias and that 43% of the records are representative of the dataset.
Page 16, lines 519-536. This comment could be placed elsewhere in the discussion of the results, linked to the evaluation of other results. Almost 95% of the records have at least the month and year. It would be interesting to reflect on whether there is also a significant representation of these records in the autumn and spring months. Are there few records in summer and winter? Is there a real over-representation of these seasons? Perhaps it could be deduced that there was a bias in the records taken in the autumn and spring months, under-representing taxa typical of the summer and winter months.
Page 17, lines 546-552. Figure S4 shows an increase in registrations around the mid-19th century in the northern cluster. It would be interesting to include if the authors have an idea of the reason for this increase (for example, the creation of a mycological society or association in the area, the work of a mycologist(s)...).
Pages 19 and 20, lines 692-696. The lower sampling in agricultural environments may have been due to a lower presence of macrofungi in this type of ecosystem compared to other biotopes. In relation to the presence of macrofungi, a favourable bias is estimated for records of this type of taxa compared to other fungi that have very small fruiting bodies or do not usually form at all (the former "Deuteromycota"). If this is the case, perhaps the authors would consider it appropriate to comment on it in the discussion of the results.
Page 20, lines 726-736. The reference to Figure 9A should be made only as “Figure 9”.
f) Tables and figures
Figure 2, Figure 3C, Figure 6A. It should be indicated what the black dots in the figures refer to. Do these dots indicate the location of the institutions or agencies that contributed records to the database?
Figure 3. The Figure 3C caption is missing. In this figure it is advisable to avoid using capital letters (e.g. lower case letters or letters of the greek alphabet).
Figure 5. It would be useful to indicate what the acronyms of abbreviations corresponds to in Figure 5B.
Table S1. It would also be advisable to indicate the acronyms of the organisations and institutions contributing to the FongiBase database.
Table S4. Please check the correct spelling of scientific names (in italics and, if appropriate, with the author(s) name(s) abbreviated).
Reviewer 2 Report
I would like to congratulate the authors for their work. In my view, these data are of paramount importance for understanding not only the biology of fungi, but their impact on human and animal lives.
Author Response
- Point by point answer to Reviewer #2:
I would like to congratulate the authors for their work. In my view, these data are of paramount importance for understanding not only the biology of fungi, but their impact on human and animal lives.
We thank the referee for this positive comment.
Reviewer 3 Report
This manuscript represents an extremely valuable collation of French fungal records. I thought that the historical context in introduction section was very useful and the analyses of the dataset provide very useful insights in particular into variations in patterns of regional recording.
I had a few general comments:-
Can the numbers of recrods, if above average, be attributed to particularly active individuals or fungus groups? If so, would be useful to provide an example (e,g some info aobut RNF/SMNF or ADONIF (from Tab S1-how many people , over what period etc.
It would be useful for readers to know what percentage of records would appear in a GBIF search.
I think this is now widely used to observe global distributions of species Did the authors think of using the more recent FungalTraits database (Polme et al.
(2021FungalTraits: a user-friendly traits database of fungi and fungus-like stramenopiles) as well as or instead of Funguild. FungalTraits has more categories and so propvides useful insights.
Not sure if it would be easy to include such an analysis.
Given the lowpercentage or records from grassland habitats but the high number of grassland species (ie. CHEGD taxa) on the IUCN Global Red List (due to habitat loss).
These species were formerly categorised as saprotrophs but it seems that they are clearly plant -associated, possibly mycorrhizal.
In FungalTraits, Hygrocybe spp Clavariacaea etc are listed as “soil_saprotroph;unspecified_symbiotroph” which is distinct from pure saprotrophs and a useful distinction.
Some more specific comments:
Fig. 1. Should legend be blow the figure? (also in other figures)
Fig2. Very useful figure but not clear what the black dots represent. In some places they obscure the relevant number. Can the dots be removed? Or possibly made smaller (plus need explanation in legend.
Fig6: See comment above re black dots
L41: these initiatives did not provide much coverage of continental areas L174/8: strange font size here
L300: Given the famous existence of AscoFrance, in the 7.8% of records for Acscomycota higher than for other coutnries (if not I guess that is due to the international reach of AscoFrance)
L350: On average,…
L533: In contrast to amateur mycologists, academic institutes such has the French Museum of National History and ADONIF make a disprortionately large contribution to the dataset
L537: patterns of mycological forays more than
L546: In the four clusters of fungal recording L558: These data suggest that
Supp L1: memory of Christian Lechat, who founded AscoFrance.
